# Synapses: The Brain’s Energy-Demanding Sites

**DOI:** 10.3390/ijms23073627

**Published:** 2022-03-26

**Authors:** Andreia Faria-Pereira, Vanessa A. Morais

**Affiliations:** Instituto de Medicina Molecular João Lobo Antunes, Faculdade de Medicina, Universidade de Lisboa, 1649-028 Lisboa, Portugal; andreiapereira@medicina.ulisboa.pt

**Keywords:** brain energy metabolism, synapses, mitochondria, glycolysis

## Abstract

The brain is one of the most energy-consuming organs in the mammalian body, and synaptic transmission is one of the major contributors. To meet these energetic requirements, the brain primarily uses glucose, which can be metabolized through glycolysis and/or mitochondrial oxidative phosphorylation. The relevance of these two energy production pathways in fulfilling energy at presynaptic terminals has been the subject of recent studies. In this review, we dissect the balance of glycolysis and oxidative phosphorylation to meet synaptic energy demands in both resting and stimulation conditions. Besides ATP output needs, mitochondria at synapse are also important for calcium buffering and regulation of reactive oxygen species. These two mitochondrial-associated pathways, once hampered, impact negatively on neuronal homeostasis and synaptic activity. Therefore, as mitochondria assume a critical role in synaptic homeostasis, it is becoming evident that the synaptic mitochondria population possesses a distinct functional fingerprint compared to other brain mitochondria. Ultimately, dysregulation of synaptic bioenergetics through glycolytic and mitochondrial dysfunctions is increasingly implicated in neurodegenerative disorders, as one of the first hallmarks in several of these diseases are synaptic energy deficits, followed by synapse degeneration.

## 1. Introduction

Even in a basal state, the human brain is one of the most energy-demanding organs. However, the brain energy expenditure is quite heterogeneous: it varies between different brain regions, such as in white vs. grey matter [1,2]; between types of neural cells, with neurons being the most energy dependent; and between type of neurons, such as excitatory vs. inhibitory [3]. Neurons have morphologically and functionally distinct subcompartments: dendrites, soma, axons and presynaptic terminals. When estimating the brain’s energy budget, synaptic transmission processes, which include postsynaptic currents, neurotransmitter recycling and presynaptic Ca^2+^ fluxes, are considered as the main energy-consuming process [4,5].

In order to fulfil these energetic needs, the brain, including synapses, uses glucose as its main substrate. For energetic purposes, glucose can be degraded through two not-incompatible pathways: glycolysis and mitochondrial respiration through oxidative phosphorylation (OxPhos). Glycolysis, which occurs in the cytosol, is independent of oxygen (O_2_) consumption and leads to the incomplete degradation of glucose into two molecules of pyruvate, which can generate two molecules of lactate. The pyruvate molecules formed in glycolysis can be imported to the mitochondria and generate acetyl-CoA molecules, which will enter the tricarboxylic acid (TCA) cycle. From this cycle, ATP, but mainly NADH and FADH_2_, electron carriers are generated and used to fulfil the OxPhos complexes chain, where, ultimately, Complex V (also known as ATP synthase) generates ATP from ADP and inorganic phosphate. OxPhos (mitochondria respiration) is able to provide high ATP yields, whereas glycolysis is a fast ATP production pathway with a lower ATP production net gain [6]. In this review, we discuss the relevance of glycolysis and OxPhos in synaptic energy metabolism, focusing on presynaptic terminals in both a resting state and under different stimulation conditions. Additionally, we discuss how distinct experimental conditions (from neuronal type, stage of development, medium use, stimulation paradigm) may affect results when studying these processes.

The importance of mitochondria at presynapses has been increasingly consolidated, and we discuss older and recent data, pointing towards mitochondria at synapses being a distinct neuronal mitochondria population, with a specific fingerprint and capability to adapt to synaptic transmission. Additionally, it is evident that mitochondria are very important organelles at synapses and their dysfunction affects synaptic homeostasis and activity, as these organelles are not only involved in ATP production, but also in calcium and reactive oxygen species regulation.

Finally, the onset of neurodegenerative disorders (NDs) is often associated with synaptic energy deficits and, ultimately, synaptic degeneration, which precedes neuronal loss (reviewed in [7,8,9]). In this review, we also point out the main glycolytic and mitochondrial dysfunctions reported on NDs, whose functional restoration has been seen as a putative therapeutic strategy to delay the onset of NDs.

## 2. Brain: The Energy Demanding Organ

Every tissue and organ in the body requires mitochondrial-derived energy, but certain tissues have especially high levels of energy demand. The brain is a clear example of such an organ. An adult brain, on average, weighs 1.4 kg and its O_2_ consumption rate is about 49 mL O_2_/min. In resting conditions, an average adult weighs 70 kg and consumes 250 mL O_2_/min, which translates to the brain representing 2% of the total body weight, but approximately 20% of the total O_2_ consumed [10,11], while skeletal muscle also consumes 20% of the total O_2_ consumed, but accounts for 40% of the total body weight [12]. These calculations were estimated in a resting and conscious state. However, it is worth mentioning that brain activity is never null, as there are always neurons firing and, therefore, the brain does not reach a real resting state. A brain’s resting state, often designated as a basal state, translates into the ongoing intrinsic activity of the brain, while in a stimulated state, subjects are engaged in task performance and/or attention-demanding tasks [13].

Additionally, it is important to distinguish between conscious and unconscious state because they impact significantly on the brain’s energy consumption. Positron emission tomography (PET) imaging showed that in an unconscious state, usually provoked by anesthesia, a great decrease in brain energy consumption is observed in comparison to individuals in a conscious state [14,15]. On the other hand, evidence has supported that under a conscious and resting state, the brain already has high baseline energetic demands that are minimally changed upon task-induced activity [13,16,17].

Importantly, it is necessary to keep in mind that these data refer to the brain as a whole and homogenous organ. However, when focusing on brain substructures, rat brain studies have shown that the white matter only consumes approximately 40% of the energy used by the grey matter [1,2,15,18]. Additionally, when comparing different neuronal types, excitatory neurons have been described as using three times more energy than inhibitory synapses [3].

### 2.1. Brain: The “Sugar” Monster

In order to fulfil these high energy demands, the brain uses mainly glucose. Glucose is a six-carbon carbohydrate whose catabolism involves ATP production reactions. In vertebrates, glucose can be catabolized through two, not-incompatible, main energy-producing pathways: glycolysis and OxPhos (Figure 1). Glycolysis occurs in the cytosol and involves sequential reactions, where it will generate, amongst other molecules, two pyruvate molecules and two ATP molecules. Typically, in the absence of O_2_, pyruvate can then be converted into lactate through lactate dehydrogenase complex (usually when LDHA isoform is prevalent). However, this conversion can also occur in aerobic conditions. Alternatively, pyruvate molecules can be imported into mitochondria, where they are converted into acetyl-CoA, which is integrated into the TCA cycle, from which two ATP molecules are produced (per glucose molecule). From this cycle, the electron carriers NADH and FADH_2_ are synthesized and can deliver electrons to the electron transport chain (ETC) Complex I and II, respectively. When electrons arrive at Complex IV, electrons and protons are conjugated with O_2_, originating H_2_O. Simultaneous to electron transfer between ETC complexes, protons accumulate in the mitochondrial intermembrane space and are used by Complex V (ATP synthase complex) to produce approximately 28 ATP molecules per glucose molecule, in a pathway designated OxPhos (Figure 1).

Glucose oxidation starts with its phosphorylation into glucose-6-phosphate (Glucose 6-P) by hexokinase. Glucose 6-P proceeds to glycolysis, ultimately forming two molecules of pyruvate per glucose molecule and yielding two ATP molecules. Of note, Glucose 6-P can also engage in the pentose phosphate pathway, important for antioxidant responses and biosynthesis pathways. Typically, in the absence of O_2_, pyruvate can then be converted into lactate through lactate dehydrogenase complex (usually when LDHA isoform is prevalent) (dotted arrows represent multiple enzymatic reactions). However, this conversion can also occur in aerobic conditions. Alternatively, pyruvate can be imported into mitochondria, where it is converted into acetyl-CoA, which is integrated into the TCA cycle, from which two ATP molecules are produced (per glucose molecule). From this cycle, electron carriers—NADH and FADH_2_—are synthesised and can deliver electrons to Complex I and II, respectively. Of note, Complex II is an integral part of the TCA cycle, catalyzing the conversion of succinate to fumarate. From electron transfer between complexes, protons accumulate in the inner membrane space and are used by Complex V to produce approximately 28 ATP molecules (per glucose molecule). Mitochondrial Ca^2+^ is thought to activate TCA cycle enzymes: NAD^+^ isocitrate dehydrogenase (conversion of isocitric acid into α-ketoglutaric acid) and 2-oxoglutarate dehydrogenase dehydrogenase (conversion of α-ketoglutaric acid into succinyl-CoA), and Complex V (Section 4.1). Commonly used glycolysis inhibitors are 2-deoxy-d-glucose (2-DG) and iodoacetic acid (IAA). 2-DG, a glucose analogue, is phosphorylated by hexokinase, resulting in 2DG-6-phosphate (2DG-6-P). However, this product cannot be further metabolized and accumulation of 2DG-6-P inhibits hexokinase, one of the initial glycolytic enzymes. IAA is an irreversible inhibitor of glyceraldehyde-3-phosphate dehydrogenase, inhibiting the following glycolytic reactions. On the other hand, oligomycin is an inhibitor of Complex V.

If a glucose molecule is completely oxidized through glycolysis, followed by OxPhos, it is estimated to consume six O_2_ molecules. Through the development of PET and functional magnetic resonance imaging (fMRI), it is possible to measure changes in blood flow, glucose and O_2_ levels and, consequently, calculate the ratio between the brain O_2_ consumption rate and the brain glucose consumption rate, also designated as an oxygen–glucose index (OGI). In recent decades, independent groups have reported OGI values from 5.03 to 5.54 [11,19,20,21] in the resting human brain, which indicates that, although a large portion of glucose uptake by the brain is fully oxidized, approximately 8% to 15% of glucose uptake is used via non-oxidative metabolism.

One of the first hypotheses to justify this discrepancy is the production of lactate. However, when calculating an OGI that included lactate (oxygen-to-carbohydrate index), it was approximately 5.6, still below the stoichiometric value of 6 [21], indicating that lactate production does not account for all non-oxidative glucose consumption.

Although glucose has been established as the main energy substrate used by the brain, other fuels have been reported to be important in distinct physiological conditions. Recently, it has been reported that, under oxygen and glucose deprivation, neuronal activity in adult bullfrogs’ brainstems was maintained constant for a few hours, however the neuronal activity resumed much earlier when glycogen phosphorylase, a rate-limiting enzyme of glycogenolysis, was inhibited [22]. Similarly, in rodent brain slices, in the absence of glucose, inhibition of glycogenolysis augmented the synaptic transmission impairments [23], suggesting that glycogenolysis might come into play when substrate availability is low. Recently, an increased focus has been given to the usage of ketone bodies as a brain energy fuel. Ketone bodies, such as β-hydroxybutyrate, acetoacetate and acetone, are converted in acetyl-CoA, fueling the TCA cycle and, later, the OxPhos pathway. Their importance has been acknowledged during starvation, as well as brain development, where it is estimated that brain uptake of ketone bodies is faster in infants than in adults [24]. Additionally, ketone bodies have been able to sustain synaptic activity in adult rats (reviewed in [25,26]). Nevertheless, the preferred cellular location for the oxidation of these alternative fuels to occur and their benefits in pathological conditions are still a matter under debate.

As noted above, in basal conditions, the brain has already very high energy demands that increase little upon task-induced activity. Focal activation of the human visual cortex estimated that basal ATP production increased from 11.1 µmol/g/min in resting conditions to 12.5–13.0 µmol/g/min upon different stimulation frequencies, translating into a relative increase of 12–17% in ATP production [17]. Additionally, it is estimated that from all ATP produced to meet stimulation demands, 98% was derived from OxPhos and 2% was derived from glycolysis [17]. A dramatic increase in lactate concentration was also observed in stimulated brain areas of rodents and humans [27,28,29,30]. This suggests that both energy pathways are enhanced upon stimulation. 

Thus, the two energy-producing pathways are not incompatible and can be accentuated at different time points after stimulation. On the other hand, in adult brains under resting conditions, different brain regions are quite heterogeneous in their ability to rely on aerobic glycolysis [31], with prefrontal cortex and lateral parietal cortex being two regions with significantly high glycolysis and the inferior temporal gyrus and the cerebellum being regions with very low glycolytic activity.

### 2.2. Synaptic Department Claims a Large Chunk of the Brain’s Energy Budget

Alongside understanding what are the major energy-producing pathways that contribute to resting and stimulated brain activity, it is important to understand what brain subcellular processes (from action potential propagation, neurotransmitter and synaptic vesicle (SV) recycling, presynaptic calcium (Ca^2+^) fluxes, postsynaptic currents and housekeeping processes) have higher energy needs.

The brain’s subcellular energy usage distribution has been investigated for decades and the brain’s energy budget was initially estimated for the rat neocortex (and, therefore, inferred to the grey matter), assuming a mean firing rate of 4 Hz [1]. This budget proposed that action potentials and postsynaptic currents are the major energy-consuming process (47% and 34%, respectively), with both neuronal and glial resting potentials consuming a smaller amount (13%) and neurotransmitter (glutamate) recycling and presynaptic Ca^2+^ fluxes accounting for an even smaller portion (6%). However, this distribution was adjusted because, unlike the unmyelinated giant axons in squid used by Laughlin and Attwell to extrapolate the energetic needs of action potentials, the vertebrate myelinated axons are energetically more efficient in action potential propagations due to a much shorter temporal window between sodium and potassium channels opening [32,33], limiting the dissipation of sodium ions and, therefore, lower energy is necessary to restore ion concentrations via sodium-potassium ATPase. Consequently, the synaptic transmission (involving postsynaptic currents, neurotransmitter recycling and presynaptic Ca^2+^ fluxes) was the major energy consumption process in the brain, accounting for 43% of the grey matter’s energy demands. Action potential propagation represented 17%, maintenance of both neuronal and glial resting potentials represented 15% and the remaining 25% were assumed for housekeeping processes (including non-signaling processes, such as organelle transport, macromolecules synthesis and degradation) [4]. In accordance, several decades ago, it was shown that most of a rat’s brain glucose consumption occurred in synapse-dense regions of the nervous system [15]. A similar approach estimated the energy budget in a rat’s optic nerve and corpus callosum, representing the white matter, at postnatal stage [2] and established that the energy use of the white matter is threefold lower than the grey matter and the maintenance of resting potentials and housekeeping processes dominated the energetic requirements of the white matter. On the other hand, the energy spent on action potentials and synaptic transmission was residual, which correlated with a significantly lower synaptic density in this region [2].

## 3. Energy Production at Presynapses

Synaptic terminals are usually localized far away from the soma. On average, in mossy fiber-associated neurons, they are distanced 2.5 cm [34] and 31 cm in basal forebrain cholinergic neurons, with an estimated length of 100 m in the human brain [35]. These distances, coupled with a low ATP diffusion rate [36], makes it imperative that synaptic energetic requirements are met *in loco*.

As synapses are the highest energy-consuming brain subcellular compartment, it is important to understand the balance required between glycolysis and OxPhos pathways in order to fulfil these synaptic energetic needs. Additionally, as presynapses are one of the most molecularly dynamic sites in the brain, it is important to understand how this balance is regulated in resting and stimulation conditions.

### 3.1. At Resting Conditions

Studies, using either luminescent or fluorescent probes to quantify presynaptic ATP levels, have shown that in postnatal rodent hippocampal neurons, presynaptic ATP levels were estimated to be 1.4 mM [37] and between 2–4 mM [38], and in both studies were shown to decrease upon synaptic activity. Curiously, when substrates like glucose were depleted, ATP levels were found constant for approximately 20 min, which might be due to the presence of pyruvate and/or the presence of transient storage of intracellular glucose. However, when glycolysis was impaired, ATP levels decreased significantly [38,39]. These levels were recovered when supplemented with a supraphysiological amount of pyruvate [38], suggesting that fueling OxPhos could compensate the loss of glycolytic ATP and maintain resting presynaptic ATP levels, extensively used for SV acidification via H^+^-vATPases [39] (Figure 2). On the other hand, glycolysis has been considered to be the major presynaptic ATP source under resting conditions (Figure 2) [37] because inhibition of early steps of glycolysis, through 2-deoxyglucose (commonly 2DG), decreased presynaptic ATP levels, whereas OxPhos inhibition, through oligomycin (Figure 1), did not change presynaptic ATP levels [37]. Nevertheless, O_2_ measurements in synaptosomes collected from different brain regions have shown an oligomycin-sensitive O_2_ consumption in non-stimulating conditions, suggestive of ATP-linked mitochondrial respiration [40,41].

The lower representation of mitochondrial ATP production in resting conditions might be related with the use of dramatically higher glucose concentrations (30 mM) than those physiologically found in rodent brains (1 mM) [42,43], which could force glucose uptake and enhance glycolysis, as shown in primary neurons [38] and isolated synaptosomes [40].

**Figure 2 ijms-23-03627-f002:**
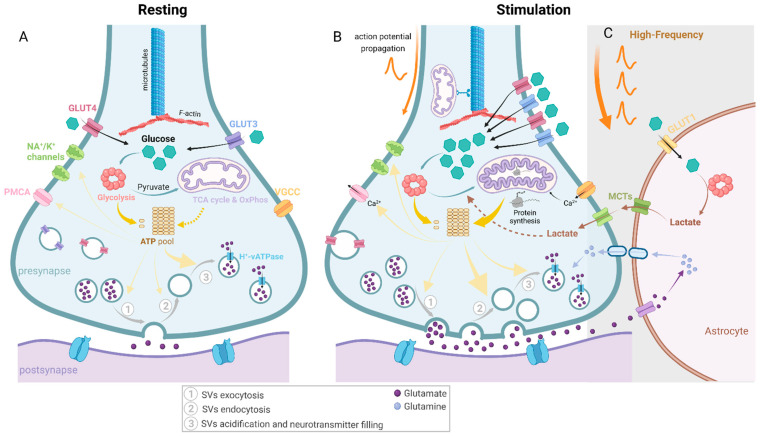
The dynamic energy requirements and fulfilments of presynapses upon distinct activation states. Glucose is the main energy substrate in the brain and given that synapses are the most energy-demanding sites in the brain, glucose degradation is important to provide ATP for synaptic pathways. Glucose is imported through glucose transporters (GLUT3 and GLUT4) and can be degraded through glycolysis in the cytosol and generate pyruvate and, ultimately, lactate. Pyruvate can be imported to mitochondria where, through multiple reactions shown in Figure 1, it will generate higher yields of ATP but at a lower rate than glycolysis [6]. Multiple reactions at presynapses require ATP, namely, synaptic vesicle recycling, endocytosis, acidification and neurotransmitter filling, priming (not represented in this figure) and exocytosis. Additionally, several ion channels in the presynaptic membrane require ATP, including sodium-potassium voltage-gated channels (Na^+^/K^+^ channels) and plasma membrane Ca^2+^-ATPase (PMCA), necessary to extrude Ca^2+^ that accumulates inside presynapses. (**A**) In a resting state, minimal energy is necessary to fuel these membrane ion channels [39], and it is speculated that due to the lack of stimuli, the energetic burden of synaptic vesicles’ (SVs) endo- and exocytosis is also reduced, despite some spontaneous release. Recently, in rat hippocampal neurons, it was suggested that SV re-acidification (through H^+^-vATPases) was the major energy consumer in presynapses (~44%) [39]. (**B**) Upon synaptic stimulation, the demand for energy is increased. Although the energy proportion required for each synaptic ATP-dependent process is not yet defined, SV endocytosis is assumed to be one of the most energy-demanding processes. Both glycolysis and OxPhos have been shown to be important to meet the activity-driven energy requirements. In active synapses, mitochondria present a higher volume and an increased cristae density [44]. Additionally, translation of both mitochondrial [45]- and nuclear-encoded mitochondrial [46] proteins have been shown to be increased upon synaptic activity. Additionally, recruitment of axonal mitochondria has also been shown to be important upon synaptic energy stress [47]. (**C**) Despite the controversy of the astrocyte-neuron lactate shuttle (ANLS) hypothesis, astrocytic lactate has been important to maintain synaptic transmission under high-frequency stimuli (HFS) [23,48,49]. Under these conditions, augmented glutamate levels in the glutamatergic synaptic clefts might boost astrocytic glutamate uptake to reconvert it back to glutamine. Concomitantly, astrocytic glycolysis is increased, generating lactate, which will be delivered to neurons through monocarboxylate transporters (MCTs) present in both neuronal and astrocytic membranes. This lactate will complement neuronal glycolysis and OxPhos in fulfilling the high SV endocytosis resultant from HFS.

### 3.2. Upon Synaptic Stimulation

Upon synaptic stimulation, an increased energy production is necessary to compensate for the activity-driven ATP usage. In 1994, a possible mechanism to explain how synaptic energetic demands are fulfilled upon synaptic activity, known as the astrocyte-neuron lactate shuttle (ANLS) hypothesis, was described [50]. According to the ANLS hypothesis, upon glutamatergic transmission, glutamate uptake by astrocytes is mainly converted into glutamine, which is used by neuronal terminals to be recycled back into glutamate. The glutamate uptake drives astrocytic glucose uptake, followed by glycolysis [50,51] and the resultant lactate is exported to neurons, where it is converted back to pyruvate to enter mitochondria and fuel OxPhos. This suggests that lactate, and not glucose, is the main neuronal substrate upon synaptic glutamatergic transmission. Indeed, compiling evidence has been showing that astrocytes and neurons have distinct molecular and metabolic profiles, indicating a higher glycolytic capacity in astrocytes and a higher OxPhos capacity in neurons (reviewed in [27,52]). Additionally, in vivo 2-photon NADH imaging in rodent hippocampus showed that, upon high-frequency stimulation (HFS), there was an initial decrease in NADH levels (NADH “dip”), which correlated with an increased OxPhos activity localized at neuronal synapses. This decrease was followed by a prolonged NADH increase (NADH “overshoot”), which correlated with increased glycolysis that was localized in astrocytes [48]. Similarly, in rat hippocampal slices, upon stimulation, a NADH dip and rapid decrease in O_2_ levels were also observed, followed by an NADH overshoot [53], mainly composed by cytosolic NADH [54]. However, when inhibiting LDH [53,54] or lactate import [54], which would impair the conversion of lactate to pyruvate by neurons, as well as pyruvate to lactate in astrocytes, O_2_ consumption and NADH increase was still observed upon synaptic stimulation, suggesting that, upon synaptic activity, the increased OxPhos does not derive from lactate import to neurons and that in this stimulation paradigm, the maintenance of the NADH increase is indicative of a neuronal glycolytic capacity.

One point of controversy in the ANLS hypothesis is provided by early and recent studies pointing towards the importance of glucose uptake and glycolysis at synaptic terminals in response to synaptic stimulation: (1) Upon stimulation, glucose uptake increases in presynaptic terminals with increased neuronal expression of Glut3 [55] and, similarly, Glut4 transporters are translocated to the presynaptic membranes [56]. Upon glucose depletion [23] or Glut4 knockdown [56], SV endocytosis is impaired. (2) Glycolytic enzymes cluster near presynaptic release sites, in *C. elegans* [57] and in rodent neurons [58]. (3) Glycolytic enzymes associate with SVs [59] and ion pumps [60,61]. These findings support that neurons, more specifically synaptic terminals, are able to perform aerobic glycolysis, which, according to the ANLS hypothesis, is mainly supported by astrocytes [50]. For further insights into the ANLS controversy, we refer the reader to other reviews [27,52,62,63].

Despite the glucose uptake and glycolysis capacity of synaptic terminals, under certain conditions, lactate still plays a crucial role. Recently, it was shown that in mouse calyx of Held synapses, upon HFS and physiological glucose (1 mM) concentration, inhibition of lactate import affected synaptic transmission [23].

Therefore, astrocytic lactate shuttle to neurons might be of higher relevance in HFS conditions [23,48] where a rapid SV recycling occurs, as it has been observed in neuromuscular junctions [64] and hippocampal synapses [65]. Additionally, HFS in rat subthalamic nucleus has been shown to increase extracellular glutamate levels [49]. Therefore, one can hypothesize that, upon HFS, augmented glutamate levels in the glutamatergic synaptic clefts may boost astrocytic glutamate uptake to reconvert to glutamine with a concomitant increase in astrocytic glucose uptake and astrocytic glycolysis, which will deliver lactate to neurons to complement neuronal glycolysis and OxPhos in fulfilling the high SV endocytosis occurring under HFS (Figure 2).

Despite the debate on the importance of astrocytic and neuronal glycolysis to fulfil energetic demands resultant from synaptic stimulation, it is generally accepted that OxPhos-driven ATP production is important to sustain synaptic stimulation in Drosophila neuromuscular junctions [66], hippocampal slices [53] and hippocampal neurons [37,38,67], as well as action potential stimulation in cortical neurons [40]. In accordance, translation of mitochondrial-encoded proteins is increased in active presynaptic hippocampal terminals [68] and in postsynaptic hippocampal terminals upon HFS [69]. Conversely, inhibition of mitochondrial translation decreased the amount of active synapses [45].

On the other hand, it has been recently emphasized that some proteins have very long half-lives, known as long-lived proteins (LLPs), which can last from days to months. Often, these LLPs are found in the brain [70,71,72] and approximately two-thirds of these LLPs are mitochondrial proteins, mainly involved in TCA cycle and OxPhos complexes. On average, the lifetime of mouse brain proteins was estimated to be 3 to 13 days, whereas the lifetime of brain mitoLLPs was estimated at approximately 28 days [72]. Interestingly, the OxPhos-related brain LLPs were found enriched in supercomplexes and in high-order assemblies of Complex V [71] and were found to be important for the assembly of these structures [70]. The formation of these supramolecular structures have been shown to be important to meet increased energy demands in other tissues [73,74,75]. Therefore, this suggests that these mitoLLPs may be important to ensure a high energetic output from mitochondria. When dissecting the brain regions where LLPs were more prevalent, proteins at synapses showed higher lifetimes than in brain homogenate [72,76]. The functional relevance of these higher protein lifetimes and what is in the synaptic molecular environment that enables their maintenance is yet to be deciphered. However, this higher prevalence of LLPs at synapses and their association with supercomplexes could constitute an adaptation to the high energy demands present at the synapse and reinforce the reliance of synapses on mitochondria.

### 3.3. Factors That Shift Synaptic Dependence on Glycolysis or OxPhos

Ultimately, the balance between glycolysis and OxPhos is not fully understood and it is important to reflect that this balance could vary due to several factors, including: (1) resting or activity state (Figure 2) (described above); (2) brain region—most studies have been focusing on hippocampal and calyx of Held synapses, however, studying different brain regions might reveal different balances between glycolysis and OxPhos, as the neuronal types and neuronal/glia composition varies across regions; (3) stage of development—it has been reported that during human brain development, in premature infants’ (estimated gestational age from 25 to 34 weeks) brains, glycolysis accounts for 100% of glucose consumption and 30% in infants (11 days to 1 year of age) [13]. Developmentally, these high glycolytic rates correlate with the highest rates of synaptogenesis and axonal and synaptic growth [77], suggesting that, besides supporting energetic demands, glycolysis is also important to support synaptic growth and formation through biosynthetic pathways [55,77]. Similarly, in the development of a rodent’s brain, lactate levels and expression of key aerobic glycolysis enzymes decline through development [78]. Therefore, it is assumed that the developing brain is considered to be predominantly glycolytic. An increase in OxPhos activity becomes evident latter on when synapses are formed and active, and the energetic demands surpass the biosynthetic needs. Indeed, OxPhos complexes II–III, IV and, even more strikingly, complex V activities in synaptic mitochondria are significantly higher in eight-week-old rats, in comparison, with animals at approximately two weeks of age [79]. Additionally, mitochondria occupancy in mouse calyx of Held increases from immature (P7) to mature (P21) stages [80], correlating with a higher mitochondria dependence upon synaptic maturation. Even though in vitro neuronal cultures increased OxPhos enzyme expression and activity throughout neuronal maturation [81], synaptic energetics studies performed in neuronal cultures used early postnatal neurons (P0–P4), which have a much higher glycolytic predisposition than adult neurons, and so could have biased the importance of the different energetic pathways. Additionally, the lifetimes of mitoLPPs in presynaptic terminals are strikingly lower in neuronal cultures than observed in brain tissue [72]. (4) Physiologically relevant medium—in vitro neuronal cultures, ex vivo slices and synaptosomes are often maintained in media that do not reflect the physiological environment present in the brain. These media often contain saturating levels of neuroactive amino acids, such as glycine, glutamate, serine, aspartate and inorganic salts, which leads to an impaired action potential generation and synaptic communication of neuronal cultures [82]. One of the most striking differences to the physiological environment is the glucose concentration, which is often applied at 25–30 mM, whereas the brain physiological concentration is 1 mM [42,43]. The use of supraphysiological glucose concentrations has been shown to increase the reliance of brain cells on glycolysis [38]. Interestingly, a recent neuronal medium developed based on brain physiological concentrations and osmolarity has been shown to positively impact on neuronal survival and synaptic activity in both human and rodent brain cellular models [82,83,84]. This importance is emphasized by distinct effects when NDUFS4 (Complex I subunit)-deficient neurons, maintained in high glucose conditions, only showed significantly lower synaptic ATP deficits and, consequently, impaired SV endocytosis when glycolysis was inhibited [38]. Therefore, it would be interesting to assess synaptic energy demands and fulfilment when using a medium mimicking the brain physiological environment.

It is worth re-emphasizing that glycolysis and OxPhos are not antagonist pathways. On the contrary, glucose-driven OxPhos requires glycolysis to occur. Mitochondria are not able ¬to directly oxidize glucose, and so, glucose initially needs to go through glycolysis, where it originates pyruvate (or lactate in astrocytes), which can be imported and completely oxidized in mitochondria. Given that the glycolysis ATP-generation reactions occur before pyruvate formation, glycolysis inhibitors (like 2DG and IAA) ultimately will reduce pyruvate formation and, therefore, will also reduce OxPhos activity, a trend observed in adipocytes for 2DG [85] and in primary cortical neurons for IAA [86]. Therefore, when comparing ATP production attained from glycolysis and OxPhos by accessing glycolysis inhibition, caution is necessary to avoid also the impairment of OxPhos.

Recently, as shown above, major focus has been employed on understanding the energetic demands on presynaptic terminals, however, postsynaptic terminals are estimated to be the synaptic compartment with the highest energy demands, based on O_2_ consumption data. According to the synaptic energy budget [53], estimated a decade ago, presynaptic action potentials represent approximately 11%; presynaptic Ca^2+^ entry, transmitter release/recycling and mGluR activity represent approximately 17%; postsynaptic currents represent approximately 46%; and postsynaptic action potentials represent the remaining 26% of the O_2_ consumed at synapses. Thus, postsynaptic processes represent almost three times more energy than presynaptic terminals, but due to the lack of studies, the balance of glycolysis and OxPhos in postsynaptic terminals is not understood. In rat hippocampal neurons, mitochondria occupy ~50% of the main dendritic shaft, but only 10–15% of filopodia and spines possessed mitochondria [87]. Interestingly, upon stimulation, mitochondria were found to accumulate and be retained in stimulated dendritic regions [87], suggesting an activity-dependent recruitment of mitochondria. Some evidence seems to support that both glycolysis and OxPhos are important for postsynaptic processes. Recently, it has been shown that OxPhos fuels postsynaptic protein translation and that the absence of mitochondria impaired spine morphological plasticity; however, in basal neuronal activity, lack of mitochondria did not impair local protein translation [88], which could be due to local ATP levels provided by glycolysis or neighboring mitochondria. Some glycolytic enzymes have been found to be highly expressed in postsynaptic densities [89] and glycolytic enzyme GAPDH has been reported to phosphorylate GABA_A_ receptors, important for inhibitory neurotransmission [90].

However, still much is unknown about the energy fulfilment, specifically at postsynaptic sites. Therefore, future studies will be important to dissect the relevance of glycolysis and OxPhos in different types of postsynaptic terminals (excitatory vs. inhibitory), between resting and activity state, as well as within the dendrite subdomains (shaft vs. spines). Interestingly, it was recently proposed that, due to the absence of mitochondria in dendritic spines and presence of MCTs in spine membranes, there could be a compartmentalization of glycolysis in the dendritic spines and OxPhos in dendritic shafts [52].

## 4. Other Mitochondrial Functions at Synapses

### 4.1. Calcium Uptake

Once action potentials, propagated alongside the axon, arrive to the presynaptic terminal, voltage-gated calcium channels open and presynaptic cytosolic Ca^2+^ levels rise. This Ca^2+^ increase can be cleared through extrusion to the extracellular space through plasma membrane Ca^2+^-ATPase (PMCA), uptake through smooth endoplasmic reticulum by Ca^2+^-ATPase (SERCA) and uptake through mitochondria via voltage-dependent anion channels (VDACs), present on the outer mitochondrial membrane (OMM), and the mitochondrial Ca^2+^ uniporter complex (MCU complex), present in the inner mitochondrial membrane (IMM) and composed of, among other proteins, mitochondrial calcium uptake 1 (MICU1) and 2 (MICU2), the Ca^2+^-sensing proteins. Overall, MCU has been reported to have a lower Ca^2+^ affinity [91]. However, this threshold seems to vary between cell types. For example, in neurons, it has been shown that the MCU complex’s threshold for Ca^2+^ intake is lower when compared with other cell types [92].

In mitochondria, Ca^2+^ is thought to activate FAD-glycerol phosphate dehydrogenase [93], pyruvate dehydrogenase phosphatase [94], NAD^+^ isocitrate dehydrogenase [95], 2-oxoglutarate dehydrogenase [95] (these last two enzymes are integrated in the TCA cycle) and Complex V [96,97] (Figure 1). Therefore, either via a direct mechanism via Complex V activation or via an indirect mechanism through indirect activation of ETC complexes, an increase in mitochondrial Ca^2+^ results in an increased capacity for ATP production. More recently, Paillard and collaborators showed that skeletal and heart mitochondria, representative OxPhos-reliant organs, have a lower MICU1/MCU expression ratio, which is translated into a lower Ca^2+^ uptake threshold to activate OxPhos [98], a mechanism that may also occur in neurons and synapses to enhance mitochondrial respiration upon synaptic transmission. Additionally, in *Drosophila* motor neurons, upon HFS, mitochondrial membrane potential decreases, followed by a sustained peak after 15 s, which is also translated in increased mitochondrial matrix pH and mitochondrial NAD(P)H levels, which were blocked upon mitochondrial Ca^2+^ uptake inhibition [99]. Consequently, at presynaptic sites, an increase in mitochondrial Ca^2+^ may couple synaptic activity and mitochondrial ATP production, placing Ca^2+^ as an important signaling molecule for mitochondria to sense the energetic needs of synaptic activity.

The calcium buffering by mitochondria is of particular importance in synapses where during high-frequency synaptic transmission, mitochondria uptake the excess Ca^2+^ and then release it slowly over time [100]. This buffering capacity is obvious in zebrafish hair synapses, where, after synaptic stimulation, the restoration of mitochondrial calcium levels takes longer than the restoration of cytosolic presynaptic Ca^2+^ levels, 5 min vs. 12 s, respectively [101]. This prevents faulty synaptic transmission, maintains the integrity of the synapse and fine-tunes cytosolic and mitochondrial Ca^2+^ signals over long periods of time (reviewed in [102]).

Additionally, in flies, MCU knockout, and, consequently, impaired mitochondrial calcium uptake, leads to decreased survival, motor impairments and deficient OxPhos activity [103]. Additionally, in primary mouse neurons, MCU knockdown accelerated SV endocytosis [104], but, interestingly, MCU pharmacological inhibition, in cortical and hippocampal neurons, has been shown to be neuroprotective in cases of excitotoxicity [105].

Besides its role in activating OxPhos, mitochondrial Ca^2+^ is also involved in mitochondrial transport. The Mitochondria Rho GTPase (Miro) protein localizes to the OMM and acts as a mitochondrial adaptor involved, alongside other proteins, in the transport and distribution of mitochondria. In neurons, it is involved in both anterograde (towards the synapse) and retrograde (towards the cell body) movement of mitochondria and has two Ca^2+^-binding EF hands. It has been reported that Miro and MCU interact and this connection is important for mitochondrial transport in axons [106]. Upon increased cytoplasmic Ca^2+^ levels, as occurs upon synaptic activity, it is hypothesized that the high Ca^2+^ levels disrupt Miro–MCU interaction and bind to the EF-hand domains of Miro, altering the chain of interactions between Miro and microtubules and arresting mitochondria (reviewed in [107]). Indeed, neurons with mutations in Miro’s EF-hand domains fail to arrest mitochondrial transport [108,109].

### 4.2. ROS Regulation

Reactive oxygen species (ROS) are chemical molecules containing, at least, one oxygen atom that, through cellular and extracellular reactions, become more reactive than molecular oxygen.

One of the most significant cellular ROS sources are mitochondria. In normal respiratory conditions, where electrons flow extensively from Complex I to Complex III via ubiquinone (Q) and ultimately to Complex IV, so-called forward electron transfer (FET), the ROS production is relatively low and is commonly derived from Complex I, Complex III and 2-oxoglutarate dehydrogenase (OGDH) enzyme, a TCA cycle enzyme that reduces NAD^+^ into NADH [110]. In these conditions, the predominant ROS source can change between tissues and cells [111].

When in high levels, ROS lead to increased oxidative damage of mitochondrial proteins and mitochondrial (mt) DNA, exacerbating mitochondrial dysfunction, which is often associated with synaptic activity impairments, neuronal loss and, consequently, neurodegeneration (reviewed in [112]).

However, low levels of ROS can serve homeostatic signaling functions, for example, increased ROS production is observed through in vitro neuronal maturation, which is correlated with enhanced OxPhos activity [81,113,114]. Additionally, mitochondrial ROS is involved in the regulation of the postsynaptic GABA receptors, strength [115], in the regulation of neuropeptide FLP-1 secretion, in *C. elegans* [116], in the modulation of dopamine release, in striatum of guinea pigs [117] and in the selection of inactive synapses for pruning during development in *Xenopus laevis* [118].

Therefore, although overall cellular ROS are often depicted as harmful molecules, in homeostatic levels, ROS can exert important signaling functions in synaptic development and plasticity. Nevertheless, the threshold between beneficial and detrimental ROS levels it is not yet understood and future studies will be important to dissect the impact on synapses homeostasis.

## 5. To Be or Not to Be a Synaptic Mitochondria?

With mitochondria playing an irrefutable role on synaptic transmission and due to the unique intracellular environment found at synapses, one could postulate that synaptic mitochondria may have “adapted” to this environment and, therefore, exhibit distinct features from mitochondria localized in other neuronal compartments.

Of note, in this review, the term synaptic mitochondria concerns mainly mitochondria localized in the presynaptic terminals, which have shown clear morphological differences to postsynaptic mitochondria in primary hippocampal neuronal cultures [119,120,121], human and mouse cortical tissues [80,122] and rat hippocampi [120,122,123] using distinct electron microscopy approaches, with presynaptic mitochondria shown to be smaller than postsynaptic mitochondria. Although not all presynaptic terminals have mitochondria in close proximity to them [124,125,126], a positive association between mitochondrial presence at synapses and synaptic longevity has been recently elucidated in adult mice [121].

Furthermore, some differences have been found between synaptic and other brain mitochondria, often referred to as non-synaptic mitochondria: susceptibility—synaptic mitochondria present an increased probability of mitochondrial permeability transition pore opening (an abrupt increase in IMM permeability solutes) [127] and a higher susceptibility to Ca^2+^ overload [128,129]; lipid membrane composition—the content of cardiolipin and ceramide is distinct between synaptic and non-synaptic mitochondria [130]; protein expression—some proteomics studies [131,132,133] have indicated protein expression differences between synaptic and non-synaptic mitochondria; TCA cycle and OxPhos activity—activity of some OxPhos complexes and TCA cycle enzymes have been found to be different in synaptic mitochondria, in comparison to non-synaptic mitochondria [79,134,135,136] and synaptic mitochondria are more sensitive to Complex I inhibition [137]; and pathology—synaptic mitochondria showed early accumulation of amyloid-β peptide in a mouse model of AD [127] and during aging, which was accompanied by premature synaptic mitochondria deficits not observed in non-synaptic mitochondria [138].

Additionally, evidence has pointed towards an adaptation of synaptic mitochondria activity and ultrastructure to meet demands arising from synaptic activity: Complex IV activity was shown to increase or decrease upon synaptic stimulation or inhibition, respectively [135]; electron tomography from calyx of Held synapses revealed that mitochondria closer to active sites, as well as rod synaptic mitochondria, have a higher cristae surface area/outer mitochondrial membrane area ratio [139], which has been linked to a higher bioenergetics capacity of these mitochondria (reviewed in [140,141]). In accordance, a recent study unraveled that synaptic mitochondria in presynaptic terminals presented enhanced cristae densities and increased cytochrome c levels compared with mitochondria in presynaptic terminals with lower activity [44].

From these evident characteristics of synaptic mitochondria, some questions arise: What defines a synaptic mitochondrion? How do synaptic mitochondria acquire their identity? 

A recent review hypothesized that synaptic mitochondria are a “differentiated” pool of mitochondria, derived from long-term maintenance of an intact mitochondria pool known as “stem” mitochondria, localized in the cell body, with a long lifespan and low metabolic activity [142]. Upon, metabolic needs, some “stem” mitochondria are selected and transported to synaptic terminals, where they become fully metabolically active and fulfil synaptic energetic requirements. This merges with evidence showing that, once damaged, synaptic mitochondria move retrogradely to be degraded in the cell body, where most of the autophagy machinery resides, to be degraded in a process denominated mitophagy [143,144]. However, it has been reported that mitophagy can occur in distal axons [145,146], as well as mtDNA replication and translation of mitochondrial-encoded proteins implicated in mitochondrial biogenesis [45,147] and reviewed in [148]. Moreover, from an energy-optimized usage point-of-view, it would not be very efficient to have “stem” and “differentiated” mitochondria constantly back-and-forth to fulfill synaptic transmission. In accordance with this, it has been shown that several mitochondrial-related proteins are translated at synapses [46], which translates into morphological and functional adaptions of already-resident synaptic mitochondria upon synaptic stimulation [44]. Additionally, upon energy stress, axonal mitochondria can be recruited to axon terminals to fulfill synaptic activity energy demands [47].

Nonetheless, the “stem-differentiated” hypothesis could be relevant for synaptic development and pruning where presynaptic terminals are still being formed and degraded, and so, recruitment of mitochondria into newly formed synaptic regions is necessary. However, it is not guaranteed that mitochondria arriving at synaptic terminals are already “differentiated” into synaptic mitochondria and are predisposed to go to presynaptic sites; mitochondrial populations at synapses might arise via a constant flux of mitochondria into axons that increasingly reach high distances. With advances in in vivo imaging to study synaptogenesis and ATP synaptic sensors, reviewed in [149], it is expected that in the next few years, some initial answers to these questions will emerge.

Electron microscopy studies performed in human cortical regions showed that the mitochondrial occupancy seems to be lower in postsynaptic terminals (5–17%) compared to presynaptic terminals (22–45%) [150] and, upon aging, hippocampal postsynaptic mitochondria are more susceptible to morphological changes than presynaptic mitochondria [123]. Nevertheless, similarly to the lack of studies assessing the balance between glycolysis and OxPhos at postsynaptic sites, is it yet to be explored whether postsynaptic mitochondria have a distinct morphological and functional fingerprint enabling them to meet the high energy demands.

## 6. When Synaptic Energy Production Fails: Neurodegenerative Diseases

Despite a wide range of clinical symptoms, ultimately it has been widely acknowledged that energy deficits are associated with neurodegenerative disorders (NDs). These energy deficits might result from impairments in glycolysis and/or mitochondrial ATP production pathways, discussed in the next sections.

### 6.1. Glycolytic Dysfunctions

In Alzheimer’s disease (AD) patients, a lower glucose uptake capacity, through low GLUT3 expression [151], is observed already in early stages [152]. This low glucose uptake is reflected in increased brain glucose levels (brain hyperglycemia), a common feature of diabetes. Interestingly, in several cohort studies, diabetes has been associated with a higher probability of developing AD [153,154,155]. On the other hand, the majority of postmortem brain studies have shown a general upregulation of glycolytic enzymes, which could be a compensation mechanism to counterbalance the decreased glucose uptake capacity (recently reviewed in [156]).

Similarly, in Parkinson’s disease (PD), PET studies in patients have also shown a decreased glucose uptake in early stages of the disease [157]. Pharmacological induction of PD in rat models has been shown to decrease glycolysis, as well as the activity of several glycolytic enzymes [158]. Interestingly, pharmacological stimulation of glycolytic enzyme PGK1, using Terazosin, in toxin-induced and genetic PD animal models increased cellular ATP levels, slowed neuronal loss, increased dopamine levels and partially restored motor function [159].

In amyotrophic lateral sclerosis (ALS), the evidence of glycolysis impairment is scarcer. The majority of PET imaging studies have shown reduced glucose metabolism in the front-parietal cortical regions and premotor cortical areas of ALS patients [160] and whole genome expression profiling in the motor cortex of sporadic ALS patients also showed significant downregulation of glycolytic genes [161]. In contrast, other studies have shown that ALS patients displayed hypermetabolism at rest and fibroblasts carrying an ALS-mutated SOD1 form have been found to increase glycolysis (reviewed in [156,162]).

Future studies will be important to specify the timing and molecular pathways impairing glucose metabolism at synaptic terminals.

### 6.2. Mitochondrial Dysfunctions

Patients with genetic disorders associated with mitochondrial functions present neurological symptoms that include epilepsy, psychomotor retardation, migraine, stroke-like episodes, dementia, peripheral neuropathy and sensitive and cerebellar ataxia [163]. Several of these symptoms are corroborated with mitochondrial impairments at synapses, such as a partial inhibition of mitochondrial Complex I and increased glutamate release, generating excitotoxicity, which might lead to epilepsy [164]. Similarly, NDUFS1-depleted *C. elegans*, a mitochondrial protein found to be mutated in Leigh syndrome, results in exaggerated presynaptic activity of acetylcholine synapses [165]. Further studies will be important to dissect the synaptic dysfunctions in animal models and mitochondrial disease patients.

On the other hand, the onset of neurodegenerative disorders (NDs) is associated with synaptic energy deficits, mitochondrial dysfunction and, ultimately, synaptic degeneration, which precedes neuronal loss [166,167], reviewed in [7,8,9].

Mitochondrial dysfunctions in NDs are often characterized by: (1) Morphological changes—which are often coupled with mitochondrial fission and fusion dysregulation. DRP1, which is involved in mitochondrial fission, has been found to be less expressed in AD postmortem brains and in primary neurons overexpressing amyloid-β (Aβ) ligands [168], and DRP1 activity has been found to be impaired in *Drosophila* and mouse neurons expressing human tau [169]. Additionally, fragmented mitochondria and DRP1-impaired activity were also observed in postmortem Huntington’s disease (HD) brains and primary neurons from transgenic mice expressing mutated human huntingtin [170]. (2) Disrupted transport—impaired anterograde mitochondrial transport (from cell body towards axon terminals) has been observed in a wide range of ND, including: amyotrophic lateral sclerosis (ALS) [171], AD [172,173,174] and PD [175,176,177,178]. Interestingly, independently of the ND studied, retrograde transport (towards the cell body) was often found to be increased in the disease [171,173,175,178]. (3) Disrupted mitophagy—mutations in mitophagy-related proteins PINK1 and PARKIN have been found in familial and sporadic forms of PD (reviewed in [179,180]). However, PINK1 mutations are also linked with energy deficits, as PINK1 deficiency has been shown to compromise Complex I activity [181]. AD patient brains and neurons expressing mutant human APP have shown enhanced Parkin-mediated mitophagy, accompanied by a decrease in cytosolic Parkin levels, suggesting a stalled mitophagy pathway, which led to aberrant accumulation of dysfunctional mitochondria [173]. (4) Reduced ATP production capacity—decreased mitochondrial membrane potential and OxPhos complexes activity have been observed in AD [135], PD [135,177,182], HD [183,184] and ALS [185,186] models. Remarkably, disrupting Complex I subunits’ expression in dopaminergic mouse neurons [187] and *Drosophila* [188] has been shown sufficient to induce progressive phenotypes resembling PD, emphasizing mitochondrial dysfunction as a central factor in PD etiology [189]. (5) Increased oxidative stress—this phenotype is usually associated with the OxPhos activity deficiencies, where deficient Complex I and III activities increase ROS production. Additionally, in NDs, increased ROS production is also accompanied by lower antioxidant activity, fostering increased ROS levels (reviewed in [112]).

Given that synaptic loss is one of the earliest hallmarks of NDs associated with energy deficits, synaptic mitochondria homeostasis plays a key role in NDs. Indeed, several studies have indicated that synaptic mitochondria dysfunction occurs well before dysfunctions in other neuronal mitochondria. For example, in an Aβ-accumulation AD model, synaptic mitochondria showed decreased mitochondrial respiration and enzymatic activity, accompanied by increased oxidative stress and compromised Ca^2+^ handling capacity in early phases of disease progression, when non-synaptic mitochondria showed no signs of impairment [127], similarly to APP transgenic rodent models [190]. Additionally, striatal synaptic mitochondria of PINK1 KO rats revealed decreased Complex I-driven respiration and proteomics data have shown a decrease in electron carrier proteins [191]. Alterations in the phospholipid composition of synaptic mitochondrial membranes [192], namely, decreased cardiolipin content, a characteristic lipid of the IMM, is involved in the assembly and stability of supercomplex structures [193] that have been shown to enhance energy production [73,74,75] and, therefore, could be contributing to a lower mitochondrial ATP production in synaptic mitochondria at early stages of NDs.

In analyzing postmortem dopaminergic neurons in PD brains [194] and brains and spinal cords from ALS patients [195], an accumulation of mitochondria with dense cristae and increased OxPhos protein levels in surviving dopaminergic and motor neurons, respectively, was observed, suggesting a compensatory energy-boosting mechanism in these NDs [195]. In line with this, several therapeutic strategies to delay the onset of NDs have been increasingly pointed towards improving and/or rescuing brain energetics by restoring oxidative phosphorylation and glycolysis and correcting mitochondrial dysfunctions and/or enhancing mitochondrial functions (reviewed in [196]).

## 7. Conclusions and Open Questions

Currently, one of the major challenges in brain and synaptic energy metabolism research is to integrate the results obtained at different resolution scales: from in vivo macrophysiological studies (using FMRI and PET imaging in adult brains) to microscopic in vitro cellular and subcellular studies (using confocal imaging, respirometry and electrophysiology assays in primary neuronal cultures and acute brain slices). As pointed out previously, these two approaches differ mainly in the developmental stage of the research model (adult brains vs. primary neurons from early postnatal days) and the use of non-physiological media, usually with saturating glucose concentrations that may bias the energetic pathways usage. Future studies should have these considerations in mind. 

Although most studies have been focusing on presynaptic energy metabolism, the synaptic energy budget estimates that, upon synaptic stimulation, 72% of the energy is used postsynaptically [53]. Recently, it has been demonstrated that mitochondria at dendrites are spatially compartmentalized and are crucial to provide energy for in loco protein translation and synaptic plasticity [88]. However, much is still to understand about the role and impact of mitochondria at postsynaptic sites, along with the main energy-consumer processes at postsynaptic sites. The development of genetic tools to probe subcompartment-specific mitochondria, ATP, Ca^2+^ (mitochondrial and cytosolic) and synaptic vesicles at different cycle stages, combined with two-photon imaging, could be very useful to help in deciphering these questions [197], in the context of both healthy and pathological conditions.

Additionally, as mitochondria at presynaptic terminals (synaptic mitochondria) possess distinct proteomics, lipidomics and morphological features from other brain mitochondria, future studies will be important to answer the following questions: How and where are these differences acquired? Are they already defined when these mitochondria are “signaled” to go to synapses? Are they acquired only upon adaption to the synaptic environment? Do these differences confer a bioenergetic advantage to synaptic mitochondria?

Deciphering the answer to these key questions will broaden our understanding on brain metabolism, in general, and the crucial role that mitochondria execute at synapses, in particular.

## Figures and Tables

**Figure 1 ijms-23-03627-f001:**
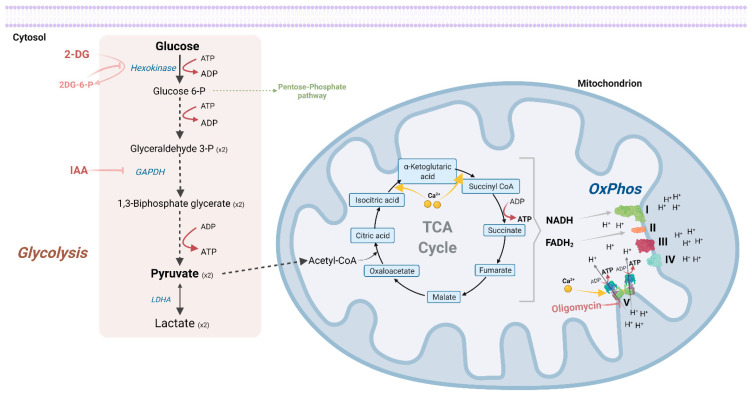
Glycolysis and oxidative phosphorylation pathways are the major ATP providers.

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
