# Peer review of "Synapses: The Brain’s Energy-Demanding Sites"

_ijms, 2022, doi:10.3390/ijms23073627_

Round 1

Reviewer 1 Report

Synapses: The brain’s energy-demanding sites

Main summary: Introduces OxPhos and glycolysis and their importance at the synapse as well as describing the most energy-expensive processes in neurotransmission. Describes how these two processes are balanced in resting and activated synaptic states. Next, the focus of the review is primarily on the role played by mitochondria at the synapse including calcium buffering and ROS formation. Differences between synaptic and non-synaptic mitochondria are described and then the failure of mitochondrial function in various neurogenerative disorders are discussed. Throughout the review and in the conclusion, the holes in our understanding of mitochondria at the synapse are discussed.

Main comments: Overall, I found a lot of the content in the review very interesting and certainly worth trying to condense into a review. My main critique of the review is that the aim and direction of the paragraphs is not clear at all. It seems as if the author just wants to talk about the role of mitochondria in energy production at the pre-synaptic as well as commenting on other important functions of mitochondria at the pre-synaptic terminal. Stating that the review aims to discuss the relationship of OxPhos and glycolysis at the synapse is a bit confusing. It also often reads as if the author is trying to determine which is “more important” for energy production at the synapse, but the two processes are merely different stages of the same processes, i.e., the product of glycolysis is the main source of fuel for the OxPhos pathway. It would make more sense to be comparing the different fuel sources for the OxPhos pathway rather than comparing the OxPhos pathway with glycolysis. They do try to point this out, but I found it a bit unclear, especially at the beginning. I wonder if it would be better to focus the review on mitochondrial energy production at the pre-synaptic terminal and just explain the different fuel sources of OxPhos as a sub-heading.

Another critique is that they fail to make it clear at the beginning that they are only talking about pre-synaptic processes in the review and merely refer to “the synapse” throughout, even when they are only talking about the pre-synaptic terminal.  

The figures seem appropriate and useful.

Abstract

Summary: The brain, and specifically neurotransmission at the synapse, requires a lot of energy. Mitochondria mediate ATP production as well as a number of other functions at the synapse. Their dysfunction contributes to neurodegeneration.

Comments: Pointing out what discussion is absent from the literature on this topic would be good as well as a sentence outlining the main points made in this review. The latter is included in the introduction however, so maybe it’s not necessary.

  1. Introduction

Summary: Brief description of glycolysis and mitochondrial oxidative phosphorylation as the main sources of energy in the brain. It is mentioned that the review will discuss the relative importance of these processes between resting and activated synapse states and how certain experimental set-ups can obscure this relationship. In addition, it states that the review will touch on how synaptic mitochondria have distinct differences from other mitochondrial populations and finally how synaptic mitochondrial dysfunction is linked to neurodegenerative disorders.

Comments: The introduction seems appropriate as it walks though all the main points to be made in the review. No “controversial” ideas were put forward. However, again to draw the reader into the review, it might be good to point out what this review contains that is lacking from other discussions in the literature.

  1. Brain: the energy demanding organ

Summary: Runs through all the usual facts about the percentage of ATP consumed by the brain, white matter, grey matter, etc., and mentions a difference in the amount of energy consumed by excitatory Vs inhibitory synapses.

Comment: Again, there is nothing “wrong” with what was said. However, as an opening to the main review body/content, it might be a bit boring as it revises the facts we all probably read about the brain in our secondary school textbooks. I probably would’ve stopped reading this review by now if I had just come across it and thought the title was interesting, or I would have just “CTRL+F ‘ed” to look for any specific information I was interested in.

2.1. Brain: the sugar monster

Summary: Briefly describes the general mechanism for glycolysis and OxPhos and then moves onto some apparent discrepancy between the amount of glucose expected to be oxidised and how much is actually oxidised. Commences the discussion around whether or not OxPhos is the main contributor of ATP upon “stimulation” and mentions how different parts of the brain are thought to rely on glycolysis and OxPhos differently.

Comments: The overview of glycolysis and OxPhos seems helpful to remind people who have forgotten their undergraduate metabolism lectures. However, I didn’t see how the oxidised glucose discrepancy was relevant to the review topic. It felt a little meandering. I would have preferred a larger discussion of the literature investigating whether glycolysis or OxPhos is more important for increased synaptic activity. In addition, it might be interesting to mention the ability of ketone body products to contribute to the OxPhos pathway in place of glucose products. Moreover, judging from Padamsey et al., 2021, while glucose is of course important for determining how much ATP is available, other pathways such as leptin signalling might regulate the amount of ATP used. Then again, this point is more post-synaptic related and the author doesn’t seem to want to delve into the post-synaptic literature.

2.2. Synaptic department claims a large chunk of the brain’s energy budget

Summary: Describes how energy-consumption is divided across the different components of synaptic transmission.

Comments: I didn’t read the paper referenced which seemed to clarify some misunderstanding in the literature about synaptic energy expenditure, however, if the paper distinguishes the fraction of energy consumed in pre- Vs post-synaptic process, it would be nice to include that.

  1. Energy production at synapses

Summary: Points out that given the distance of synapses from the soma and ATP’s low diffusion rate, it is important to understand how energy is supplied quickly & directly to synapses.

3.1. At resting conditions

Summary: Discusses the role of glycolysis/glucose/OxPhos in pre-synaptic resting conditions briefly.

3.2. Upon synaptic stimulation

Summary: Moves onto how ATP is generated at the synapse when it is activated. Describes the astrocyte-neuron-lactate-shuttle and the for/against arguments of this theory. Next, some examples of changes in mitochondrial dynamics with synaptic activation are given. A link between the distribution of long-lived mitochondrial and synaptic proteins is also mentioned.

Next (Line 338 – 458), the failings in the literature to address this question in different brain regions, different stages of development, post-synaptic energy usage Vs pre-synaptic and other holes in different experimental set-ups, e.g. cell culture are discussed.

Comments: The points made in this section are all very interesting. It should have been made clearer earlier that the author was primarily referring to pre-synaptic terminals. Line 338 – 458 should be a separate section with a separate subheading.

  1. Other mitochondrial functions at synapses

4.1. Calcium uptake

Summary: Discusses a few examples of mitochondria being involved with calcium buffering at the synapse and how calcium influx can trigger OxPhos. Briefly mentions the role of calcium in halting mitochondrial transport as well.

Comments: This section seems fine and relevant when tied back to the relationship of calcium and increased ATP production. It was slightly meandering, and it might be good to include a closing statement about how the relationship between mitochondria and calcium signalling highlights further the significant role of mitochondria in activity-related ATP production.

4.2. ROS regulation

Summary: Describes the capacity of mitochondria to produce ROS and mentions a link between ROS production and OxPhos.

Comments: I’m not sure how relevant this is given that the review is supposed to be about how different ATP production pathways are balanced to mediate synaptic activity. Maybe it could just be included as a couple of sections when talking about how impaired energy production at the synapse leads to neurodegeneration.

To be or not to be a synaptic mitochondria

Summary: Discuss some of the differences that have been discovered between synaptic and non-synaptic mitochondria and how this might arise.

Comments: I really like this section, but I am not sure if the direction of the review is making sense anymore. It should be rearranged with the aim of the review clearly highlighted at the beginning.

When synaptic energy production fails: neurodegenerative diseases

Summary: Describes the usual literature concerning mitochondrial failure in neurodegeneration.

Comments: There’s a lot of uncertainty about what is actually happening in these disorders with regard to mitochondria. I feel like less people would be aware of mitochondrialopathies which are known to entail specific mitochondrial mutations. It might be interesting to highlight the effects of these conditions and if any of the symptoms/disturbed mechanisms seen in these disorders demonstrate further how important mitochondria are in synapse function and energy production.

Conclusions and Open Questions

Summary: Goes back over the points already made in the review concerning holes in the literature and some important questions yet to be asked.

Comments: Again, perfectly good conclusion but I think the original direction of this review meandered and judging by the conclusion, etc., what the author wants to address and focus on is the role of mitochondria in pre-synaptic energy consumption and function as well as some other interesting questions to be asked about mitochondria & the synapse. The aims stated at the beginning about deciphering some balance between OxPhos and glycolysis at the beginning are confusing. They are certainly relevant and helpful for the readers background knowledge, but overall, I think that those sections could be condensed a bit and reorganised such that the author’s real aim of discussing the role of mitochondria at the pre-synaptic terminal is clearer.

Reviewer 2 Report

In this review, Faria-Pereira and Morais offer a nice discussion of the bioenergetic strategies adopted at synapses, with a focus on a resting state and upon synaptic transmission. This work focuses on glycolysis and mitochondrial oxidative phosphorylation acting specifically at synapses, believed to be the main energy demanding substructures in the brain. As neurons have functionally and energetically heterogenous compartments, this synapse-centered view of the topic is key and of great benefit to both basic and clinical researchers in the field. I therefore believe that this is a solid and relevant work that deserves publication as long as some minor aspects are addressed (please see below). In addition, my main concern is the poor punctuation, which makes the text of difficult reading and sometimes misleading. Careful attention should be taken (if possible by a native speaker).

Minor points:

  • Lines 25-35: In the framework of discussing the degree of energy expenditure by different brain processes, it would be relevant to also include the updated work by Howarth et al Attwell 2012 (Journal of Cerebral blood flow & metabolism).

  • Lines 84-92: The rationale that the authors use to conclude that the high energetic demand of the grey matter derives from excitatory synapses is weak and should be revisited. The grey matter comprises “cell bodies, synapses, dendrites and unmyelinated axons” and not only excitatory synapses. Reference 3 describes differences in energy consumption between different neuronal types that likely arise from the relative numbers of excitatory/inhibitory cells, neuronal size and spiking activity; and most likely do not express the energy used by excitatory vs. inhibitory synapses. A stronger argument should be sought.

  • Lines 120-128: As this review focuses on energy production and not alternate pathways using glucose, I feel that the information herein contained is irrelevant and may blur the content of this section. I would therefore propose to remove it.

  • Lines 129-142: Rewriting of this paragraph is needed to avoid misunderstandings. Reference 15 shows that the higher energy demand upon visual stimulation is met through OxPhos, as it is clearly written on lines 141-142. However, following the description of reference 15, authors conclude that “Whether this increased ATP production is mainly derived from OxPhos or glycolysis is still a matter of debate.”. I understand that the arguments used to favor “greater glycolytic impact” are valid; but in my opinion, it should be clearly stated that in the paradigm of task-induced brain activity used on reference 15 energy requirements were met by The arguments favoring glycolysis in different organisms/brain regions should be discussed thereafter.

  • Line 204: The discussed literature on section 3 “Energy production at synapses” mainly concerns the presynaptic terminal. This should be clearly mentioned at the beginning of this section.

  • Line 206: Please replace reference 29, which is a book, for more primary papers.

  • Lines 217-218: The delayed drop on ATP levels upon glucose depletion may be due to cellular storage of glucose that maintains glycolysis functional for a short 20 min period of time. Discussing this possibility is important, as this finding does not discard the requirement of glycolysis at resting conditions.

  • Lines 221-222: The authors conclusion “suggesting that OxPhos is important to maintain resting pre-synaptic ATP levels” is misleading as rescue of ATP levels by pyruvate could just be a compensatory effect and not a direct demonstration that OxPhos is required. This is actually in apparent disagreement with the findings described just after (lines 222-226). Please rewrite as to soften this conclusion.

  • Line 223: Reference 34 concerns dendritic mitochondria and does not describe changes in presynaptic ATP levels. Please correct and cite the appropriate studies.

  • Lines 213-229 and figure 2: From my understanding of the literature described in this section, proof that OxPhos is needed to keep presynaptic ATP levels at resting conditions is lacking. I suggest either describing in more detail direct proof of this or drawing the arrow linking “TCA cycle & OxPhosp” to “ATP pool” in figure 2A as a dashed line.

  • Lines 248-249: Please include references supporting the described findings.

  • Line 262: Reference 31 demonstrates that neuronal activity drives both glycolysis and mitochondrial function. It would be relevant to mention this study in section 3.2.

  • Line 269: Please cite studies supporting that “The glutamate uptake drives astrocytic glucose uptake followed by glycolysis”.

  • Lines 282-285: Authors describe studies showing that inhibition of lactate production does not seem to affect metabolic changes upon synaptic stimulation. It would be great if the authors could elaborate a bit more on what this may suggest.

  • Lines 311-336: For simplicity, this paragraph would benefit if it is cut in two: one part on involvement of OxPhos and another on the LLPs.

  • Line 338: Please define “these two energy production pathways”. Is it glycolysis vs Oxphos? Or astrocytic vs neuronal?

  • Lines 345-346: the terms “premature infants” and “infants” are vague. Could the authors be more specific in the age of these infants?

  • Lines 405-407: Could the authors elaborate on what they believe are the reasons (e.g. technical limitations, etc) for the lack of knowledge on “energy fulfilment at postsynaptic sites” as compared to presynaptic sites? And which directions should be followed to explore so.

  • Lines 421-423: Please include references supporting the described findings. Also, it would be informative to indicate in figure 1 at which step the mentioned enzymes act.

  • Line 428: Please define MICU1 and provide brief explanation of its function.

  • Lines 440-442: Readers would benefit from a more detailed explanation of the benefits attributed to mitochondria calcium buffering capacity.

  • Line 449: Please define OMM.

  • Line 453: Please define EF.

  • Lines 448-458: It is difficult to reconcile how mitochondrial calcium uptake is involved in mitochondrial transport as Miro is located on the OMM and so theoretically exposed to cytosolic and not mitochondrial [Ca]. Please provide better explanation.

  • Line 504: “opening” of which channels? Please be more specific if possible.

  • Lines 502-526: To my understanding these paragraphs concern only presynaptic mitochondria. It may be confusing to define “non-synaptic mitochondria” as the pool of mitochondria not located at presynaptic terminals, as these can also be present at spines. I would suggest a different terminology like non-presynaptic mitochondria or other mitochondria. Also, it is curious that presynaptic mitochondria display characteristic features that optimize their bioenergetic capacity; which is, from my understanding of section 5, not the case of postsynaptic mitochondria. Considering that postsynaptic sites spend 3 times more energy than presynaptic terminals (lines 399-400), this observation comes as a surprise. It would be relevant if the authors could discuss this aspect. Is this due to lack of studies focusing on postsynaptic mitochondria? Or could it be due to different spatiotemporal energetic needs of pre/postsynapses; different basal levels of glycolysis, etc?

  • Section 6: In addition to the described impairments in OxPhos and mitochondria in neurodegenerative disorders, authors should mention whether changes in the glycolytic pathway have been described or not.
